

# A decision support system for primary headache developed through machine learning

Fangfang Liu, Guanshui Bao, Mengxia Yan and Guiming Lin

Shanghai Jiao Tong University, School of Medicine, Shanghai Ninth People's Hospital, Shanghai, Huangpuqu, China

## ABSTRACT

**Background**. Primary headache is a disorder with a high incidence and low diagnostic accuracy; the incidence of migraine and tension-type headache ranks first among primary headaches. Artificial intelligence (AI) decision support systems have shown great potential in the medical field. Therefore, we attempt to use machine learning to build a clinical decision-making system for primary headaches.

**Methods**. The demographic data and headache characteristics of 173 patients were collected by questionnaires. Decision tree, random forest, gradient boosting algorithm and support vector machine (SVM) models were used to construct a discriminant model and a confusion matrix was used to calculate the evaluation indicators of the models. Furthermore, we have carried out feature selection through univariate statistical analysis and machine learning.

**Results**. In the models, the accuracy, F1 score were calculated through the confusion matrix. The logistic regression model has the best discrimination effect, with the accuracy reaching 0.84 and the area under the ROC curve also being the largest at 0.90. Furthermore, we identified the most important factors for distinguishing the two disorders through statistical analysis and machine learning: nausea/vomiting and photophobia/phonophobia. These two factors represent potential independent factors for the identification of migraines and tension-type headaches, with the accuracy reaching 0.74 and the area under the ROC curve being at 0.74.

**Conclusions**. Applying machine learning to the decision-making system for primary headaches can achieve a high diagnostic accuracy. Among them, the discrimination effect obtained by the integrated algorithm is significantly better than that of a single learner. Second, nausea/vomiting, photophobia/phonophobia may be the most important factors for distinguishing migraine from tension-type headaches.

Corresponding author
Guanshui Bao, baogs@163.com

## INTRODUCTION

Headache is one of the most common symptoms in neurology clinics. More than 90% of the general population reports suffering from headache during any given year, which can be regarded as a lifetime history of head pain (*Hagen et al., 2018*). In China, the 1-year prevalence of primary headache is reported to be 23.8%. The prevalence of migraine
was 9.3%, and that of tension-type headaches was 10.3% (*Yu et al., 2012*). Due to the massive population base, patients spend 672.7 billion yuan each year because of primary headaches, accounting for 2.24% of China's GDP (*Yao et al., 2019*). Although headaches do not seriously threaten the lives of patients, they can severely affect their work and quality of life, causing them to withdraw from society, and place heavy burdens on the patients' psychology, physiology and the families of patients as well as China's national economy (*Takeshima et al., 2019*; *Saylor & Steiner, 2018*; *Malmberg-Ceder et al., 2019*).

Headaches are divided into primary headaches and secondary headaches. There are many causes of headaches. Due to the similarity of symptoms, it is easy for general practitioners to miss or misdiagnose the type of headache. Furthermore, the International Headache Society (IHS) released the latest headache classification in January 2018, which is the International Classification of Headache Disorders (ICHD-III) (*Headache Classification Committee of the International Headache Society , 2018*), which lists more than 200 headache variants. This complicated classification creates a very challenging task for general clinicians. There is no objective gold standard, which contributes to the difficulty of diagnosing and classifying headaches. In addition, because the medical community has generally not paid enough attention to headaches in clinical practice for a long time, the proficiency level of clinicians regarding the headache classification is uneven. For example, "vascular headache" and "nervous headache" are still used to diagnose primary headache.

Thus, much progress remains to be made toward standardizing and improving the accuracy of the clinical diagnosis of headache.

According to reports, primary headaches occur more frequently than secondary headaches, and the incidence of migraine and tension-type headache ranks first among the types of primary headache (*Guerrero et al., 2011*). Migraines include migraines with aura and migraines without aura. Migraines without aura are typically unilateral, pulsating, and moderate to severe headaches; daily physical activity can exacerbate these headaches, and they are often accompanied by nausea/vomiting and/or photophobia/phonophobia. Aura is the gradual appearance of visual, sensory, or other central nervous system symptoms before or during the headache. Tension-type headaches are the most common type of primary headache; attacks of this type of headache are not frequent and usually last several minutes to several days. These headaches are typically characterized by mild to moderate bilateral compression or band-like sensation; they are not aggravated by daily physical activity and are not often accompanied by nausea/vomiting, or photophobia/phonophobia. Although there are large differences between typical migraines and tension-type headaches, the symptoms of most patients are not typical, especially in cases of tension-type headache and migraine without aura. Thus, it is often difficult to distinguish between them. Due to the many differences in the treatment of the two disorders, misdiagnosis and missed diagnosis inevitably delay the appropriate treatment of the patients (*Porter et al., 2019*).

At present, the development of artificial intelligence (AI) is in full swing. Automatic classifiers, which are faster than clinicians due to their ability to analyze massive amounts of medical data, can minimize errors in disease recognition and improve diagnostic accuracy. Support vector machine (SVM) models, random forests, etc. have been used in the diagnosis of heart disease (*Krittanawong et al., 2020*), breast cancer (*Huang et al.,*

*2017*), prostate cancer (*Li et al., 2018*), Alzheimer's disease (*Shen et al., 2018*), and many other diseases. The future of AI in neurology is promising, with potential applications ranging from the prediction of outcomes of seizure disorder (*Abbasi & Goldenholz, 2019*), the grading of brain tumors (*Kocher et al., 2020*), the upskilling of neurosurgical procedures (*Senders et al., 2018*), and the rehabilitation of stroke patients to the use of smartphone apps for monitoring patient symptoms and progress (*Chae et al., 2020*).

For the proper recognition of headache, high-quality computer software could be very useful. As early as 2013, *Krawczyk et al. (2013)* proposed the automatic diagnosis of primary headaches through machine learning. The comparison of diagnostic performance between the advanced machine learning technology and clinicians revealed that the computer decision support system achieved a higher diagnostic accuracy. More recently, *Vandewiele et al. (2018)* proposed an end-to-end decision support system to improve the efficiency of diagnosis and follow-up in the treatment of primary headaches. The decision support system includes three large components and a shared backend: a mobile application for patients, a web application for doctors to visualize the collected data, and an automatic diagnosis module. In the automatic diagnosis module, a decision tree is used for modeling (*Vandewiele et al., 2018*). *Xiangyong (2019)* proposed a primary headache decision-making system based on international headache diagnostic criteria and conducted a four-month clinical evaluation at the International Headache Center of a tertiary hospital in Beijing. Good results have been obtained in terms of the sensitivity and specificity of this system for diagnosing headaches (*Xiangyong, 2019*). Considering the incomplete language rules when human experts express their knowledge, *Khayamnia et al. (2019)* improved the algorithm and used the Learning-From-Examples (LEF) algorithm to train the diagnostic fuzzy system, and the correct recognition rate reached 85%. They further proposed SVM- and multilayer perceptron (MLP)–based decision support systems, which achieved accuracy rates of 90% and 88%, respectively (*Khayamnia et al., 2019*). *Simi'c et al. (2021)* create a hybrid intelligent system for diagnosing primary headache disorders, applying various mathematical, statistical and artificial intelligence techniques. Although various types of research have been devoted to computer decision support systems, there are still major obstacles to their widespread use in clinical practice. Machine learning applied to medical records can be an effective tool to predict disease. In China, machine learning methods for diagnosing primary headache remain lacking.

Therefore, to achieve a higher headache diagnostic accuracy, we collected information from primary headache patients in neurology clinics through questionnaires and then entered the data into the system. We compared various machine learning algorithms to identify the best model. Furthermore, through feature selection, we identified the most important factors for distinguishing migraines from tension-type headaches, which provide a basis for clinicians to quickly diagnose headaches.

## MATERIALS & METHODS

This is a cross-sectional study designed to obtain a diagnostic discriminant model for migraines and tension-type headaches and to screen out the most important factors for

distinguishing the two. The study was approved by the Ethics Committee of the Ninth People's Hospital affiliated to Shanghai Jiao Tong University Medicine (approval no. SH9H-2021-T72-1), and met the requirements of the Declaration of Helsinki. Eligible patients were patients diagnosed with headaches between October 2019 and September 2020 at the Department of Neurology, Shanghai Ninth People's Hospital. All the patients were residents of China. Before the study, we obtained signed informed consent from the participating patients. Two weeks after a patient's questionnaire was collected, we followed up on the patient's headache improvement to further verify the diagnosis. Finally, we included 173 patients with a definite diagnosis of primary headache (84 patients with migraine headaches and 89 patients with tension-type headaches) for research.

## Data acquisition

First, we designed a paper questionnaire for the outpatients to complete. The questionnaire included a total of 19 questions to collect the demographic data (age, sex, occupation, height, and weight) on the patients and their headache characteristics (course, duration, nature, location, severe intensity, accompanying symptoms, triggers, alleviative methods, and whether activity aggravates the headache). After analysis and modification of the questionnaire by three experienced neurologists, the questionnaire was deemed effective for collecting patient-related information, and the data obtained were reliable to a certain extent.

Furthermore, information on related examinations and MRI were used to rule out the patient's secondary factors. Three neurologists were invited to make a diagnosis for each patient based on the questionnaire information we collected. Based on both the diagnosis and the follow-up results, each patient was accurately diagnosed. Due to the low proportion of primary headaches such as neuralgia and cluster headaches among the collected observations, we excluded these rare types of headaches to reduce the problems caused by sample imbalance. Ultimately, 173 patients (84 patients with migraines and 89 patients with tension-type headaches) were included in the study (Fig. 1). Each patient's headache may have had multiple natures or been accompanied by multiple symptoms. Therefore, we performed a binary classification of the collected data and obtained a total of 48 variables. Considering that the incidence of many variables was extremely low, we first identified 10 variables with statistically significant differences between migraines and tension-type headaches. After data transformation and data reduction, the data sheet used to acquire data during the clinical interview is shown in Table 1.

## Discriminant model establishment

Using the above 10 feature variables, we randomly divided the entire dataset into a training set and a test set at several ratio variations (60:40, 70:30, 80:20) and used holdout and cross-validation methods to build the primary headache discriminant models. Data analysis was performed in Python (version 3.6.1). We used the decision tree, random forest, gradient boosting, logistic regression, and SVM algorithms to construct discriminant models.
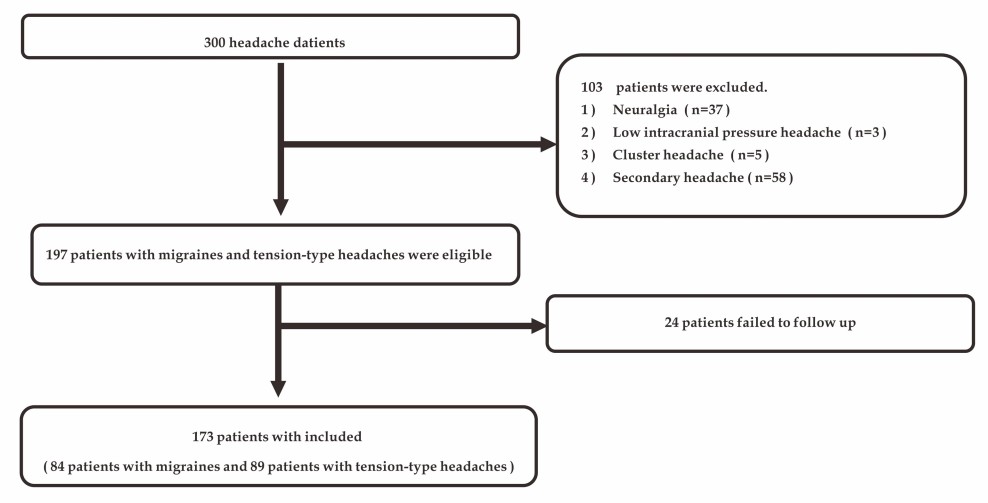

**Figure 1    Study flow chart.**

### Decision tree

Decision tree is a nonparametric supervised learning method. The basic idea is to separate binary variables and construct a tree that can be used to predict the category of new variables. It traverses the training data and condenses the information into the internal nodes and leaf nodes. Firstly, it summarizes decision rules from a series of data with features and labels, then present these rules in a tree structure to solve classification problems.

### Random forest

Random forest is an integrated algorithm that completes the learning task by constructing and combining multiple learners. These learners are always classification trees. Firstly, the data is classified by all trees, then the new category is determined by the majority decision principle. It is nonparametrically interpretable and compatible with many types of data, with high prediction accuracy.

### SVM

SVM is a binary supervised classification method, which shows many unique advantages in solving small sample, nonlinear and high-dimensional pattern problems. The purpose of this method is to find an optimal decision boundary in a multidimensional space, which can maximize the distance between two closest points in different categories. This method can process various types of data. From an academic point of view, SVM may be the closest machine learning algorithm to deep learning.

### Gradient boosting

Gradient boosting is another integrated algorithm. Like random forest, it constructs multiple learners and brings them together into a final summed prediction. The main advantage of this method is that can process various types of data flexibly, including continuous values and discrete values.

**Table 1  Patient baseline characteristics.**

| Characteristics | Migraine (n = 84) | Tension-type headache (n = 89) | Total | P-value |
|---|---|---|---|---|
| Sex/n(%) | – | – | – | – |
| Female | 20(23.8) | 39(43.8) | 59(34.1) | P = 0.01 |
| Male | 64(76.2) | 50(56.2) | 114(65.9) | |
| Course/n(%) | | | | |
| Year | 11(13.1%) | 38(42.7%) | 49(28.3) | P < 0.001 |
| Month | 73(86.9%) | 51(57.3%) | 114(65.9) | |
| Throbbing/n (%) | | | | |
| Yes | 17(20.2) | 6(6.7) | 23(13.3) | P = 0.01 |
| No | 67(79.8) | 83(93.3) | 150(86.7) | |
| Occiput/n (%) | | | | |
| Yes | 22(26.2) | 43(48.3) | 65(37.6) | P = 0.00 |
| No | 62(73.8) | 46(51.7) | 108(62.4) | |
| Severe intensity/n (%) | | | | |
| Light | 13(15.5) | 30(33.7) | 43(24.9) | |
| Medium | 44(52.4) | 51(57.3) | 95(54.9) | P < 0.001 |
| Heavy | 27(32.1) | 8(9.0) | 35(20.2) | |
| Nausea/vomiting/n (%) | | | | |
| Yes | 44(52.4) | 16(18.0) | 60(34.7) | P < 0.001 |
| No | 40(47.6) | 73(82.0) | 113(65.3) | |
| Photophobia/phonophobia /n (%) | | | | |
| Yes | 27(32.1) | 4(4.5) | 31(17.9) | P < 0.001 |
| No | 57(67.9) | 85(95.5) | 142(82.1) | |
| Spark/n (%) | | | | |
| Yes | 11(13.1) | 3(3.4) | 14(8.1) | P = 0.02 |
| No | 73(86.9) | 86(96.6) | 159(91.9) | |
| Change after activities/n (%) | | | | |
| Aggravate | 41(48.8) | 18(20.2) | 59(34.1) | |
| Unchanged | 38(45.2) | 62(69.7) | 100(57.8) | P < 0.001 |
| Relieve | 5(6.0) | 9(10.1) | 14(8.1) | |
| Alleviative methods/n (%) | | | | |
| Persistence | 9(10.7) | 14(15.7) | 23(13.3) | |
| Rest | 25(29.8) | 45(50.6) | 70(40.5) | P = 0.00 |
| Drug | 48(57.1) | 25(28.1) | 73(42.2) | |
| Else | 2(2.4) | 5(5.6) | 7(4.0) | |

### Logistic regression

Logistic regression is a supervised learning algorithm to solve the binary classification problem, which is used to estimate the probability of a certain category. It also can process various types of data.

Furthermore, we combined the accuracy and F1 score as evaluation indicators of the model through the common confusion matrix, and then measured the prediction result

(receiver operating characteristic, ROC) curve and the area under the ROC curve. The F1 score is the harmonic mean of the precision and recall. It is used in statistics to measure the accuracy of two classifications and assume that recall and precision are equally important.

$$F1score = \frac{2Precision*Recall}{Precision+Recall}.$$

### Feature selection

The ten variables have redundancies in terms of allowing clinicians to quickly distinguish whether a headache is a migraine or tension-type headache. Therefore, we identified the two variables that are most meaningful for diagnosing migraines and tension-type headaches through feature ranking. First, we adopted traditional univariate biometric analysis and then performed machine learning analysis. For the univariate test, we used the Pearson correlation coefficient (PCC) (*Xiangyong, 2019*), and the chi-square test to compare each feature between the two groups. The PCC represents the linear correlation between the elements of the two lists. If the elements in the two lists are linearly correlated, the absolute value of the PCC will produce a high value close to 1; otherwise, it will be close to 0. The chi-square test is applied to two features to observe the probability of the distribution occurring by chance. Each feature tested will produce a *p*-value. Although the *P*-value does not represent the strength of the relationship between the two variables, it provides an indication: the lower the *p*-value is, the greater certainty that the two variables are related. Furthermore, we ranked the feature importance with the random forest method. The random forest model is a nonlinear decision tree combination model. It is easy to implement and has superior performance. It was once known as "the method that represents the level of integrated learning technology". Using the random forest algorithm for feature selection is superior to the use of linear discriminant analysis and mean squared error methods for eliminating redundant features. The main idea is to judge how much each feature contributes to each tree in the random forest and then to take the average value and evaluate the contribution of each feature separately. Compared with the PCC, the random forest is more capable of mining the deep correlation of data features.

Afterwards, in a similar way we did before, we decided to investigate how the predictive power would behave when using only the two selected features.

## RESULTS

### Patient baseline characteristics

In our study, we enrolled 300 patients with primary headache. A total of 103 patients were excluded according to the exclusion criteria. In addition, 24 patients were not followed up within 2 weeks (Fig. 1). Finally, we included 173 patients (84 patients with migraines and 89 patients with tension-type headaches). We randomly divided the data from these 173 patients into a training set and test set at several ratio variations (60:40, 70:30, 80:20). Our questionnaire collected information on 48 patient characteristics through 19 questions. We used the chi-square test to identify 10 informative characteristics and included them in the study (Table 1).

**Table 2  Evaluation of the discriminant effect of various models.**

| | 80:20 | | | 70:30 | | | 60:40 | | | Mean | | |
|---|---|---|---|---|---|---|---|---|---|---|---|---|
| | Accuracy | F1 | AUC | Accuracy | F1 | AUC | Accuracy | F1 | AUC | Accuracy | F1 | AUC |
| Decision tree | 0.74 | 0.69 | 0.74 | 0.74 | 0.65 | 0.64 | 0.64 | 0.69 | 0.78 | 0.72 | 0.68 | 0.72 |
| Random Forests | 0.89 | 0.86 | 0.90 | 0.90 | 0.78 | 0.79 | 0.79 | 0.74 | 0.85 | 0.80 | 0.79 | 0.85 |
| Gradient boosting | 0.89 | 0.87 | 0.91 | 0.91 | 0.71 | 0.70 | 0.70 | 0.79 | 0.86 | 0.79 | 0.79 | 0.82 |
| Logistic regression | 0.91 | 0.90 | 0.95 | 0.95 | 0.82 | 0.88 | 0.88 | 0.77 | 0.87 | 0.84 | 0.83 | 0.90 |
| SVM-linear | 0.89 | 0.87 | 0.84 | 0.84 | 0.81 | 0.82 | 0.82 | 0.75 | 0.81 | 0.82 | 0.81 | 0.82 |

## Model building

For the above 10 feature variables, we used the decision tree, random forest, gradient boosting, logistic regression, and SVM algorithms to construct the discriminant models. After the cross-validation, the mean accuracy, F1 score were calculated through the confusion matrix (Table 2), the discrimination result curve (ROC curve) was constructed, and the area under the ROC curve were measured. The mean accuracy of the decision tree is 0.72, which was significantly lower than that of the integrated learning algorithm and SVM models. The random forest, gradient boosting algorithm, and SVM models have similar discrimination effects; their mean accuracy scores were 0.80, 0.79, and 0.82, and the mean areas under the ROC curves were 0.85, 0.82, and 0.82, respectively and the mean F1 score were 0.79, 0.79, and 0.81, respectively. Logistic regression had the best discrimination effect, with the mean accuracy reaching 0.84 and the mean area under the ROC curve also being the largest among the methods, at 0.90. The discrimination effect achieved by the integrated algorithm was better than that of a single learner method, and among the models, logistic regression achieved the best discrimination effect.

## Feature selection

For feature selection, we applied two methods: univariate statistical analysis and machine learning. For the univariate test, we used the PCC (Fig. 2) and the chi-square test (Table 3) to compare each feature between the two groups and rank them according to *p*-value. Through the univariate chi-square tests, we determined that the smallest *p*-values were obtained for the variables indicating whether the headache was accompanied by nausea/vomiting and whether the headache was accompanied by photophobia/phonophobia. These two variables have the greatest power in distinguishing the two disorders. The PCC confirmed the strong correlation between elements of the two lists. The odds ratios (ORs) for nausea/vomiting and photophobia/phonophobia were 0.4, and were higher than those of the other headache-ralated variables. Through a simple correlation analysis, we observe that patients with nausea/vomiting or photophobia/phonophobia were more likely to be diagnosed with migraine headache than tension-type headache. To confirm and explore the deeper relationship between the two disorders, we obtained the feature importance rankings through the random forest model (Table 4). Among the variables, nausea/vomiting and photophobia/phonophobia had importance values of 0.1897 and 0.1573, respectively, ranking them as the top two variables. To verify the predictive power of nausea/vomiting and photophobia/phonophobia, we trained the logistic regression on these two features,
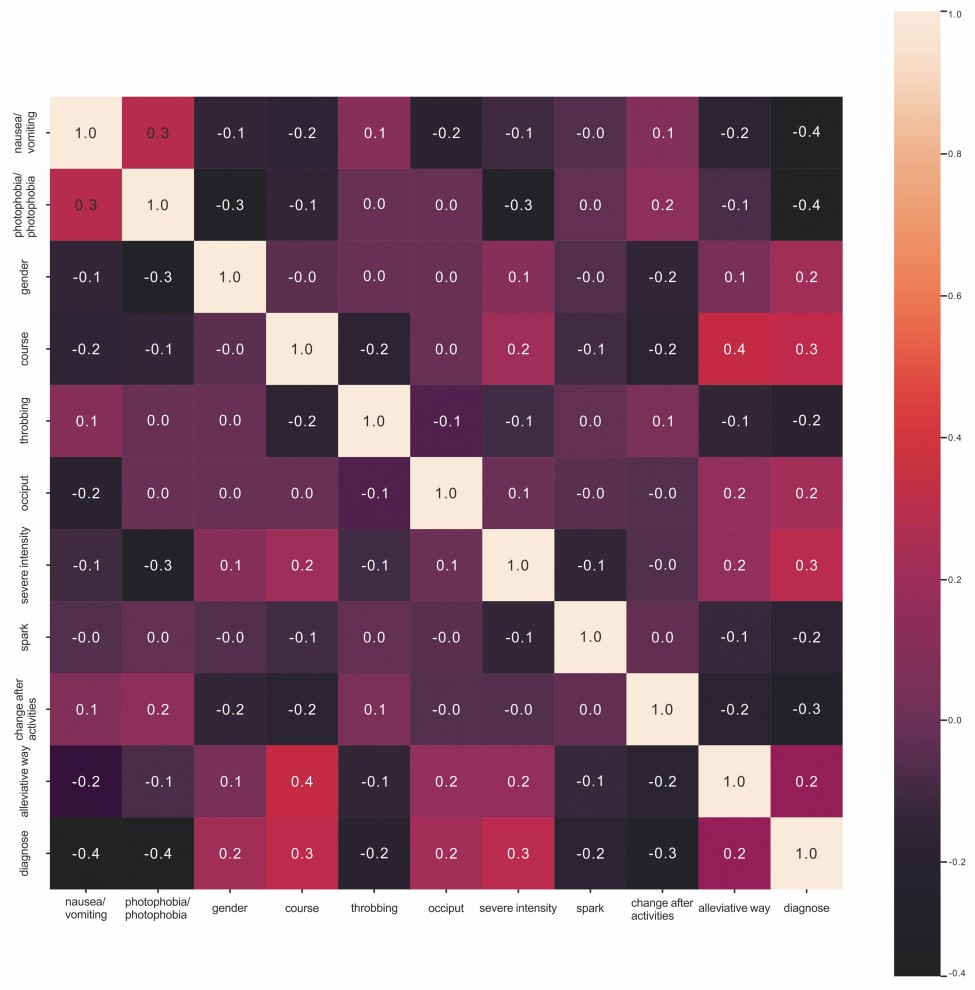

**Figure 2** Pearson correlation coefficient.

with the mean accuracy reaching 0.74 and the mean area under the ROC curve reaching 0.74 (Table 5).

In clinical practice, compared with patients with tension-type headaches, migraine patients have more severe headaches and longer disease courses, and their headaches are usually accompanied by nausea/vomiting and photophobia/phonophobia. In contrast, tension-type headaches are generally mild, and not accompanied by nausea/vomiting and photophobia/phonophobia. Our results are consistent with clinical experience. Therefore, we further compared the headache severity and nausea/vomiting and photophobia/phonophobia between the two types of patients (Fig. 3). Compared with patients with tension-type headaches, migraine patients were more likely to experience nausea/ vomiting and photophobia/phonophobia. Migraines were more severe and were mainly distributed among the moderate to severe cases, while tension-type headaches were mainly distributed among the mild to moderate cases.

**Table 3  Chi-square test.**

| Characteristic variable | *P*-value |
|---|---|
| Photophobia/phonophobia | $P < 0.001$ |
| Nausea/vomiting | $P < 0.001$ |
| Course | $P < 0.001$ |
| Change after activities | $P < 0.001$ |
| Severe intensity | $P < 0.001$ |
| Alleviative way | $P = 0.00$ |
| Occiput | $P = 0.00$ |
| Throbbing | $P = 0.01$ |
| Spark | $P = 0.02$ |

**Table 4  Random forest importance ranking.**

| Characteristic variable | Importance |
|---|---|
| Nausea/vomiting | 0.1897 |
| Photophobia/phonophobia | 0.1573 |
| Change after activities | 0.1144 |
| Course | 0.1124 |
| Severe intensity | 0.1083 |
| Alleviative way | 0.0837 |
| Occiput | 0.0754 |
| Spark | 0.0604 |
| Throbbing | 0.0444 |

**Table 5  Evaluation of the predictive power of the two selected features.**

| Logistic regression | Accuracy | F1-score | ROC-AUC |
|---|---|---|---|
| 80:20 | 0.74 | 0.61 | 0.71 |
| 70:30 | 0.71 | 0.69 | 0.73 |
| 60:40 | 0.76 | 0.74 | 0.78 |
| Mean | 0.74 | 0.68 | 0.74 |

## DISCUSSION

### Model building

AI is being applied to all types of fields, and its application to the medical field is a way for us to follow this trend. We used machine learning to identify primary headaches, which provided a starting point for advancing the transformation of AI. In this study, we established a discriminant model for the two types of primary headaches (migraine and tension-type headache) by machine learning algorithms based on 10 indicators. The diagnosis of primary headache, which is a functional disorder without an objective gold standard for diagnosis, is very difficult. Especially for the intermediate state of these two diseases, the ICHD-III diagnostic criteria are suitable for the diagnosis of only typical

**Figure 3 The correlation between headache severe intensity, nausea/vomiting, and photophobia/phonophobia.**

headache. For atypical headache and the intermediate headache state, many clinicians can rely only on their own clinical experience, and this subjective approach inevitably has a great impact on the accuracy of disease diagnosis. In other words, clinical diagnoses made by clinicians are highly subjective, varied and inconsistent. Furthermore, some scholars believe that there may be overlap of multiple primary headaches, where multiple headache symptoms exist simultaneously. Such overlapping headaches are common in cases of migraine and tension-type headache. In addition, there are treatment differences among the different types of headaches. Only clear diagnoses can improve these treatments. This intermediate headache state and the overlapping conditions make it difficult for clinicians to accurately diagnose primary headaches. Previous studies on primary headaches have been focused mainly on expert decision-making systems based on international diagnostic standards (*Costabile et al., 2020*; *Roesch et al., 2020*; *Hui et al., 2018*). However, it is difficult to make a diagnosis based on the ICHD-III criteria for the intermediate state and the overlap of clinical diseases. Perhaps it would be more efficient and effective to diagnose diseases through individualized learning and reasoning based on samples than via a pure expert decision-making system. Machine learning methods are an attractive option for such a task because they offer fast, precise and intelligent analysis of multidimensional data. Therefore, in this study, we constructed a model through different machine learning algorithms and explore the differences between samples. In addition, for related headache data, it is possible to perform cluster analysis and improve headache classification. Because of the subjective nature of the diagnosis, perform their evaluations independently and reach different conclusions for the same case. After the promotion and application of the decision-making system and through continuous learning and revision, the diagnostic criteria used by clinicians can develop in the same direction.

## Feature selection

To help clinicians quickly grasp the focus of the disease, the 10 variables were screened through univariate statistical analysis and machine learning to identify the most important factors for distinguishing migraines and tension-type headaches. The two most important factors were nausea/vomiting and photophobia/phonophobia. They represent potential independent predictors. In previous studies on simplified headache diagnostic criteria (*Martin et al., 2005*), a univariate migraine model including nausea achieved a positive likelihood ratio of 4.8 and a negative likelihood ratio of 0.23. By including the three variables

for nausea, photophobia, and throbbing headache, the migraine model achieved a positive likelihood ratio of 6.7 and a negative likelihood ratio of 0.23. The ID Migraine™ screening instrument has been found to be an effective and reliable migraine screening instrument, among the possible variables, disability, nausea, and photophobia provide the best performance (*Lipton et al., 2003*). In our research, although we did not separately screen for nausea, vomiting, photophobia, and phonophobia, the results we obtained through statistical analysis and machine learning are generally consistent with those of previous studies. To ensure the integrity of the experiment, we trained the logistic regression based on these two features. According to the results, the multi-features model is better than the two-features model. However, the two features selected can help clinicians grasp the focus of the disease as soon as possible. Nausea/vomiting, photophobia/phonophobia, and phonophobia play a vital role in distinguishing migraines from tension-type headaches.

Inevitably, our study has flaws. First, our discriminant model includes only the two types of headaches with the highest incidence: migraine and tension-type headache. Although the model can solve most of the problems related to the clinical diagnosis of headaches, other primary headaches and secondary headaches are not included. Therefore, adding other headache categories will be a future direction of expansion of our system. Second, the diagnosis of headache is strongly affected by the clinical experience of the clinician. Although we followed up with each patient after 2 weeks to assess headache improvement and verify the diagnosis, changes in the patient's living habits or other factors might have impacted on the follow-up results. Third, we included headache patients who visited a doctor, leading to selection bias. Patients with mild headaches who did not seek medical attention from a doctor were not included in the study. Finally, our sample size was small, we need to increase the sample size to verify and test the model.

## CONCLUSIONS

Primary headache is a disorder with high incidence and low diagnostic accuracy. The goal of this research is focused on the design and implementation of decision support system for diagnosing primary headaches. This study used machine learning to construct a discriminant model for migraines and tension-type headaches. The discriminant effect achieved by the integrated algorithms, such as the random forest and gradient boosting algorithms, was better than that of a single learner approaches, and the logistic regression model achieved the best discrimination effect. Further research could be focused on creating new and more efficient tools and systems to help and improve physicians' work and make diagnoses better. In addition, we identified the most important factors for the identification of the two diseases through statistical analysis and machine learning: nausea/vomiting and photophobia/phonophobia. These two factors represent potential independent factors for identifying migraines and tension-type headaches, which can help clinicians quickly grasp the focus of headaches. However, our sample size was small, and we need to increase the sample size to verify and improve the model.

### Funding

This work was supported by the Jinhua Science and Technology Bureau (No. 2020-3-036), and the project of the Shanghai Science and Technology Commission (14411972200). The funders had no role in study design, data collection and analysis, decision to publish, or preparation of the manuscript.

### Grant Disclosures

The following grant information was disclosed by the authors:
Jinhua Science and Technology Bureau: 2020-3-036.
Shanghai Science and Technology Commission: 14411972200.

### Competing Interests

The authors declare there are no competing interests.

### Author Contributions

- Fangfang Liu conceived and designed the experiments, performed the experiments, analyzed the data, prepared figures and/or tables, authored or reviewed drafts of the paper, and approved the final draft.
- Guanshui Bao conceived and designed the experiments, authored or reviewed drafts of the paper, and approved the final draft.
- Mengxia Yan and Guiming Lin analyzed the data, authored or reviewed drafts of the paper, and approved the final draft.

### Human Ethics

The following information was supplied relating to ethical approvals (i.e., approving body and any reference numbers):

The study was approved by the Ethics Committee of the Ninth People's Hospital, Shanghai Jiao Tong University School of Medicine (approval no. SH9H-2021-T72-1).

### Clinical Trial Ethics

The following information was supplied relating to ethical approvals (i.e., approving body and any reference numbers):

The study was approved by the Ethics Committee of the Ninth People's Hospital affiliated to Shanghai Jiao Tong University Medicine (approval no.SH9H-2021-T72-1).

### Data Availability

The raw measurements are available in the Supplementary File.

### Clinical Trial Registration

The following information was supplied regarding Clinical Trial registration:

ChiCTR2100044029

## Supplemental Information

Supplemental information for this article can be found online at http://dx.doi.org/10.7717/peerj.12743#supplemental-information.

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
