# Peer review of "A decision support system for primary headache developed through machine learning"

_PeerJ, doi:10.7717/peerj.12743_

## Round 0.1 · original submission · Major Revisions

· Academic Editor

Major Revisions

Major revisions are needed.Please revise and resubmit after meticulous revision, as the same peer reviewers will re-review the revised manuscript.

Reviewer 1 ·

Basic reporting

Introduction
• This article relates to the implementation of machine learning techniques in classifying types of headaches (migraine / tension-type headaches).
• Five well-known machine learning techniques were implemented to serve the classification purpose.

Originality
• The authors use the primary dataset to build the model.
• Although the techniques used (machine learning techniques) are relatively standard, the dataset's source provides some originality to indicate novelty.
• References – only 36% of the cited papers/articles are published from the year 2017 onwards. I recommend that more latest references should be added.

Language
• This article needs to be proofread. There are some typo / grammatical mistakes—for example, feature selection instead of “feature selecting” (line 179). I suggest you have a colleague who is proficient in English and familiar with the subject matter review your manuscript or contact a professional editing service.

Experimental design

• Overall, the scientific rigour of this article is pretty basic. There some missing elements are needed to improve the robustness of the results.
• Are data pre-processing methods (e.g. data cleaning, transformation, reduction) were used or did I miss something fundamental here?
• Principal components analysis (PCA) is one of the data reduction approaches that could be used to improve this model.
• Five ML techniques were chosen as classifiers (lines 154-156). Despite the usefulness of these techniques, I recommend the authors explain the technical justification (related to the data attributes, data types and types of classes (binary-like classification). The provided explanation (lines 157-170) only covers overall ideas of those techniques' behaviour.
• As the distribution of the classes is almost equal (84 (migraines):89 (tension-type)), the classification accuracy rate also can be used together with other evaluation metrics. The results (as in Table 2) show the proof of the results.
• Although F1-score is a good measurement, it works well when it comes to the balance between precision and recall. I did not find any solid justification for relying on F1-score as the dataset classes ratio is pretty balanced.
• Are there comparisons were made based on selected features? (e.g. top two variables vs ten variables (that was selected from the total of 48 variables). This comparison is essential to see the compactness of the model according to the Occam’s Razor perspectives (models should have as few parameters/features as possible).

Validity of the findings

• Model evaluation techniques need to be improved – e.g. adding more methods. (as the dataset = 173 records), it requires more intensive evaluation constructs. As reported in line 153, it seems 80:20 (training vs testing ratio) is not sufficient to show the model's generalisation power.
• Recommendation – to conduct the experiments based on several ratio variations (60:40, 70:30, 80:20), using holdout and cross-validation methods. This part is crucial to evaluate the real classification power of each classifier.

Additional comments

• The flow of this article could be improved by explaining the process using the machine learning reporting approach (data requirement, collection, understanding, preparation, modelling, and evaluation).
• The font style for tables and the clarity of the diagram can be improved.

Reviewer 2 ·

Basic reporting

Please state your method flow clearly from questionnaire to feature selection to classification. In line 24 which is your method(s), should you remove your findings and put it in the results section. Rephrase the sentence and add the missing flows/parts of your method.

Why did you include North America cases in your introduction paper? What are the items that you want to compare with China and where is the reference? [citation needed]

In line 66, computer scientist mostly do not know that migraine has two types. Please rephrase / elaborate more on that. "They" in line 66 refers to? Consider use the proper word for it.

In line 80, consider elaborate more on your area of AI related topic rather than generally speaking about it. [citation needed when elaborate]

In line 84, tell us how automatic classifier can become faster than the clinicians? In what way? Rephrase.

In line 105, in what way current practice that include ML in headache diagnostic are having major obstacle in clinical practice? How can you conclude/tell us more on your 0.9 F1-score is better than 90% accuracy rate in line 105?

Are your trying to introduce another method of assessing headache by using questionnaire compared to the other research that you have cited?

Experimental design

In line 135, please elaborate more on the experts contribution towards making the questionnaire to a certain extent.

Please rephrase line 147 to 152. Reduce the grammatical errors.

More grammatical errors in line 165.

Elaborate more in line 166 to 167. Please specify in which area related to biological information?

Please include formula on how do you calculate the F1-score, precision and recall. Why does it important to evaluate your findings?

In line 179, feature selection or feature selecting? or selecting feature?

Please rephrase line 180.

Please put your reference in line 199 to 200. Figure / table / something?

Validity of the findings

No comment

Additional comments

Please elaborate your claim in line 267 to 269. Avoid making general statement in discussion as it does not reflect or discussing your method/findings.

In line 269 to 270, what are you trying to tell here? Please discuss further on the topic.

Starting line 300, please consider to remove numbering in explaining the flaws.

Please revise the conclusion part.

The acknowledgement should include your research funder or the experts who helps in designing and revising the questionnaire.

---

## Round 0.2 · Minor Revisions

· Academic Editor

Minor Revisions

Dear Authors,Minor revisions required.Please do them soonest.Thank you.

Reviewer 1 ·

Basic reporting

The flow of the paper is still moderate. Perhaps, it could be improved a bit

Experimental design

My initial comment: Are there comparisons were made based on selected features? (e.g. top two variables vs ten variables (that was selected from the total of 48 variables). This comparison is essential to see the compactness of the model according to the Occam’s Razor perspectives (models should have as few parameters/features as possible).

The authors’ Answer: We did not make comparison based on selected features. In clinical work, important clinical characteristics can help clinicians quickly grasp the focus of the disease, but in the process of disease diagnosis, the fewer reference characteristics, the easier it is to make misdiagnosis. Relying on two variables to diagnose headaches is inadequate in clinical practice. Therefore, to make sure a higher headache diagnostic accuracy, clinicians should collect as much relevant information as possible.
*update: I agree with the justification. However, within machine learning (e.g. feature engineering), the analysis of important features are equally important to maintain the classification power by minimizing the trade-off between performances over simplicity.

* Update: I agree with the justification. However, within machine learning (e.g. feature engineering), the analysis of important features are equally important to maintain the classification power by minimizing the trade-off between performances over simplicity.

Validity of the findings

My initial comment: Model evaluation techniques need to be improved – e.g. adding more methods. (as the dataset = 173 records), it requires more intensive evaluation constructs. As reported in line 153, it seems 80:20 (training vs testing ratio) is not sufficient to model's generalisation power. Recommendation – to conduct the experiments based on several ratio variations (60:40, 70:30, 80:20), using holdout and cross-validation methods. This part is crucial to evaluate the real classification power of each classifier.
Authors’ Answer: Consider Reviewer'sviewer’s suggestion, we have conducted the experiments based on several ratio variations (60:40, 70:30, 80:20). The final result has been also corrected in the revised manuscript.

*Update: The authors added the recommended experiments. However, no comparison table and explanation were made to justify the selection of the best training: testing ratio. I am expecting some tables to show the results.

---

## Round 0.3 · accepted · Accept

· Academic Editor

Accept

Congratulations your revised article has been accepted.

Reviewer 1 ·

Basic reporting

No comment

Experimental design

No comment

Validity of the findings

No comment

Additional comments

The author has amended the article accordingly—my final suggestion, to have a proper check to improve the writing flow.